# Cytochalasans from the Endophytic Fungus *Aspergillus* sp. LE2: Their Structures, Antibacterial and NO Production Inhibitory Activities

**DOI:** 10.3390/jof9030374

**Published:** 2023-03-19

**Authors:** Yong-Chao Li, Jing Yang, Yuan-Zeng Zhao, Qin-Di Ma, Ben-Guo Liu, Jun-Liang Sun

**Affiliations:** 1School of Life Science and Technology, Henan Institute of Science and Technology, Xinxiang 453003, China; 2Henan International Joint Laboratory of Plant Genetic Improvement and Soil Remediation, Xinxiang 453003, China; 3School of Food Science, Henan Institute of Science and Technology, Xinxiang 453003, China

**Keywords:** *Aspergillus* sp. LE2, endophytic fungus, isolation and structure elucidation, antibacterial activity, NO production inhibition

## Abstract

Six new cytochalasans—namely, aspergicytochalasins A–F (**1**–**6**)—together with five known analogs were isolated and characterized from the endophytic fungus *Aspergillus* sp. from the medicinal plant *Lonicera japonica*. The structures of the new compounds were established by NMR and MS methods as well as single crystal X-ray diffractions. Compounds **3** and **4** showed weak antibacterial activities to *Staphylococcus aureus*, with MIC values of 128 and 64 μg/mL, respectively. Compounds **1**, **3**, **5** and **6** showed inhibitory activities on NO production, with IC_50_ values less than 40 μM.

## 1. Introduction

Endophytic fungi have established special functions that can produce a number of bioactive metabolites due to the process of co-evolution with host plants in a long history; these metabolites are widely used in healthcare and pharmaceutical areas [1]. Therefore, the discovery of bioactive substances and even lead compounds from endophytic fungi has become one of the important contents in the study of natural products of fungi.

*Lonicera japonica* Thunb. is a famous medicinal plant with a wide range of pharmacological activities, including antibacterial, anti-inflammatory, antiviral and anticancer, which has been utilized in most East Asian countries [2]. Its dried bud or the initial blooming flower, called honeysuckle, is used as a tea for daily healthcare in China [3]. As a widely used medicinal herb, *L. japonica* has been extensively studied on its chemical composition, which has led to identification of a number of flavonoids, iridoids, phenolic acids and saponins with a wide range of biological activities [4,5,6,7,8,9,10,11]. However, the endophytic fungi system of *L. japonica* and the chemical components have not been studied. Several endophytic strains were isolated from the plant of *L. japonica*, which exhibited potential plant growth improvement [12,13,14,15]. Therefore, it is necessary to carry out a systematical study on the endophytic fungi of *L. japonica* and their metabolites.

In our study, sixteen fungal species were identified from both the stem and flower of *L. japonica*. One of the strains, identified as *Aspergillus* sp. LE2, exhibited certain antibacterial and anti-inflammatory activities; therefore, we carried out a chemical study on its secondary metabolites. In the current study, six previously undescribed cytochalasans—namely, aspergicytochalasins A–F (**1**–**6**)—together with five known analogs, were obtained from the cultural broth of *Aspergillus* sp. LE2 (Figure 1). The structures of the new compounds were identified. The new compounds were evaluated for their antibacterial activity against *Escherichia coli* and *Staphylococcus aureus* as well as for their anti-inflammatory activities by inhibiting nitric oxide (NO) production in LPS-induced RAW 264.7 macrophages. Herein, the isolation, structure elucidation and their biological activities are reported.

## 2. Materials and Methods

### 2.1. General Experimental Procedures

To acquire the infrared (IR) spectra, we used a Shimadzu Fourier Transform Infrared spectrometer with KBr pellets. The ultraviolet (UV) spectra were acquired using a double beam spectrophotometer (UH5300). The optical rotations were acquired using a Rudolph Autopol IV polarimeter (Hackettstown, NJ, USA). High-resolution electrospray ionization mass spectra (HRESIMS) were obtained using an Agilent 6200 Q-TOF MS system or a Thermo Scientific Q Exactive Orbitrap MS system. Nuclear magnetic resonance (NMR) spectra were obtained using a Bruker Avance III 600 MHz spectrometer (Bruker, Karlsruhe, Germany). Column chromatography (CC) was performed using a Sephadex LH-20 (GE Healthcare) with silica gel (80–100 and 200–300 mesh) and RP-18 gel (20–45 μm; FuJi). High-performance liquid chromatography (HPLC) was performed using an Agilent 1260 liquid chromatography system equipped with Zorbax SB-C18 columns (5 μm; 9.4 mm × 150 mm or 21.2 mm × 150 mm).

### 2.2. Fungal Material

The fungus *Aspergillus* sp. LE2 was isolated from the fresh stems and flowers of *Lonicera japonica* Thunb. that were collected from Xinxiang, Henan Province, China. The fungal species was identified by ITS sequencing with the accession number KM051392.1. A voucher specimen (LJEF20210513-LE2) was deposited at the Department of Life Science and Technology, Henan Institute of Science and Technology.

### 2.3. Fermentation, Extraction and Isolation

The rice culture medium consisted of 5% yeast, 6% glucose, 0.15% pork peptone, 0.04% KH_2_PO_4_ and 0.06% MgSO_4_. The initial pH was adjusted to 6.5.

This fungal strain was cultured in a potato dextrose agar medium at 24 °C for 7 days to be used as the seed. The seed was then transferred to a rice medium and incubated at 20 °C in a dark place for 30 days. The rice medium in each bottle (500 mL) contained rice (70 g) and water (70 mL). A total of 50 bottles were used in this work.

The solid rice culture broth (3.5 kg) was extracted five times with EtOAc to obtain a crude extract (92 g). The crude extract was separated by CC over silica gel (80–100 mesh) to obtain six fractions (A–F), as eluted with a solvent system of CHCl_3_/MeOH (from 1:0 to 0:1 *v*/*v*). Fraction B (12 g) was separated by MPLC over RP-18 silica gel eluted with MeOH/H_2_O (from 0:100 to 100:0) to obtain five subfractions, B1–B5. Fraction B2 (1.8 g) was separated by CC over silica gel (200–300 mesh) eluted with CHCl_3_/acetone (10:1) to obtain six subfractions, B2_1_–B2_6_. Fraction B2_3_ (320 mg) was purified by prep-HPLC eluted with CH_3_CN/H_2_O (from 20:80 to 60:40 for 30 min) to obtain compounds **4** (1.8 mg) and **5** (6.3 mg). Fraction B2_4_ (400 mg) was purified by prep-HPLC eluted with CH_3_CN/H_2_O (from 20:80 to 60:40 for 30 min) to produce **2** (2.6 mg) and **6** (4.5 mg). Fraction B2_5_ (900 mg) was further separated by CC over silica gel (200–300 mesh) eluted with petroleum ether/acetone (7:1) to obtain **9** (12 mg), **10** (7 mg) and a mixture. The mixture was dissolved in MeOH to obtain crystal **8** (330 mg). Fraction D (16 g) was fractionated by MPLC over RP-18 silica gel eluted with MeOH/H_2_O (from 0:100 to 100:0) to obtain 12 subfractions (D1–D12). Fraction D3 (800 mg) was separated by CC over silica gel (200–300 mesh) eluted with CHCl_3_/acetone (12:1) to obtain compound **6** (4.5 mg) and a mixture. The colorless crystal **7** (19 mg) was deposited from the mixture. Fraction D5 (1.2 g) was separated by CC (silica gel, 200–300 mesh) to produce compounds **3** (3.8 mg) and **1** (3.1 mg), as eluted with petroleum ether/acetone (about 8:1).

*Aspergicytochalasin A* (**1**): colorless crystals (MeOH), mp: 218–220 °C; ${\left[\alpha \right]}_{25}^{D}$ + 76.4 (*c* 0.12, MeOH); UV (MeOH) *λ*_max_ (log *ε*) 248 (3.01) and 210 (3.76) nm; IR (KBr) *ν*_max_ 3423, 1688, 1651, 1277, 1102, 1016, 974 and 703 cm^−1^; ^1^H- (600 MHz) and ^13^C-NMR (150 MHz) data (methanol-*d*_4_), see Table 1; HRESIMS *m*/*z* 502.25627 [M + Na]^+^ (calcd for C_29_H_37_NO_5_Na^+^: 502.25639).

*Aspergicytochalasin B* (**2**): colorless oil; ${\left[\alpha \right]}_{25}^{D}$ + 88.8 (*c* 0.64, MeOH); UV (MeOH) λ_max_ (log *ε*) 210 (3.85) nm; ^1^H-NMR (600 MHz) and ^13^C-NMR (150 MHz) data (methanol-*d*_4_), see Table 1; HR-ESI-MS *m*/*z* 518.25122 [M + Na]^+^ (calcd for C_29_H_37_NO_6_Na^+^: 518.25131).

*Aspergicytochalasin C* (**3**): colorless oil; ${\left[\alpha \right]}_{25}^{D}\;$+ 39.3 (*c* 0.18, MeOH); UV (MeOH) *λ*_max_ (log *ε*) 250 (3.02) and 210 (3.76) nm; ^1^H- (600 MHz) and ^13^C-NMR (150 MHz) data (methanol-*d*_4_), see Table 1; HRESIMS *m*/*z* 516.23566 [M + Na]^+^ (calcd for C_29_H_35_NO_6_Na^+^: 516.23566).

*Aspergicytochalasin D* (**4**): colorless oil; ${\left[\alpha \right]}_{25}^{D}\;$+ 25.6 (*c* 0.21, MeOH); ^1^H- (600 MHz) and ^13^C-NMR (150 MHz) data (methanol-*d*_4_), see Table 2; HRESIMS *m*/*z* 518.25116 [M + Na]^+^ (calcd for C_29_H_37_NO_6_Na^+^: 518.25131).

*Aspergicytochalasin E* (**5**): colorless oil; ${\left[\alpha \right]}_{25}^{D}$ + 100.7 (*c* 0.23, MeOH); ^1^H- (600 MHz) and ^13^C-NMR (150 MHz) data (methanol-*d*_4_), see Table 2; HRESIMS *m*/*z* 496.26947 [M + H]^+^ (calcd for C_29_H_38_NO_6_^+^: 496.26936).

*Aspergicytochalasin F* (**6**): colorless oil; ${\left[\alpha \right]}_{25}^{D}$ − 34.0 (*c* 0.14, MeOH); ^1^H- (600 MHz) and ^13^C-NMR (150 MHz) data (methanol-*d*_4_), see Table 2; HRESIMS *m*/*z* 520.26679 [M + Na]^+^ (calcd for C_29_H_39_NO_6_Na^+^: 520.26696).

*X-ray Crystallographic Data for aspergicytochalasin A (**1**)*: C_29_H_37_NO_5_·H_2_O, *M* = 497.61. Cell constants of *a* = 9.8035(6) Å, *b* = 12.4869(8) Å, *c* = 11.0745(7) Å, *α* = 90.00°, *β* = 98.951(2)°, *γ* = 90.00°, *V* = 1339.18(15) Å^3^ and *T* = 150(2) K. Space group *P*1 21 1, Z = 2 and *μ*(CuKα) = 1.54178. Goodness of fit: 1.054. Flack parameter: 0.08(4). CCDC number: 2240408 (data are available at https://www.ccdc.cam.ac.uk, accessed on 12 February 2023). For details of the crystallographic data, see the Appendix A.

### 2.4. Antibacterial Assay

Bacterial strains *Escherichia coli* ATCC25922 and *Staphylococcus aureus* subsp. *aureus* ATCC29213, purchased from the China General Microbiological Culture Collection Center, were cultured in Mueller Hinton broth (MHB) (Guangdong Huankai Microbial Sci. &Tech. Co., Ltd., Guangzhou, China) at 37 °C overnight with shaking at 200 rpm. The assay was carried out by following the method reported in the literature [16]. Penicillin G sodium salt (Biosharp) was used as the positive inhibitor control.

### 2.5. Nitric Oxide Production in RAW 264.7 Macrophages

This assay was carried out by following the method reported in the literature [17]. Murine monocytic RAW 264.7 macrophages and a RPMI-1640 medium were acquired from HyClone. Griess reagents, including reagent A and reagent B, were purchased from Sigma. The absorbance of samples was measured at 570 nm with a 2104 Envision multilabel plate reader (Perkin-Elmer Life Sciences, Inc., Boston, MA, USA). S-methylisothiourea sulfate (SMS) was used as a positive control.

## 3. Results and Discussion

Compound **1** was isolated as colorless crystals (MeOH). Its molecular formula was identified as C_29_H_37_NO_5_ by HRESIMS (measured at *m*/*z* 502.25627 [M + Na]^+^; calcd for C_29_H_37_NO_5_Na^+^: 502.25639), corresponding with 12 degrees of unsaturation. In the ^1^H-NMR spectrum (Table 1), five olefinic protons at *δ*_H_ 7.14 (2H, d, *J* = 7.4 Hz, H-2ʹ and H-6ʹ), 7.26 (2H, t, *J* = 7.4 Hz, H-3ʹ and H-5ʹ) and 7.18 (1H, t, *J* = 7.4 Hz, H-4ʹ) revealed the presence of a mono-substituted phenyl group. Four olefinic protons at *δ*_H_ 5.86 (1H, dd, *J* = 15.0, 10.1 Hz, H-13), 5.29 (1H, m, H-14), 5.63 (1H, d, *J* = 15.5 Hz, H-22) and 7.02 (1H, m, H-21) indicated two *E*-form double bonds whereas the protons at *δ*_H_ 5.30 and 5.08 (each 1H, s, H_2_-12) suggested one terminal double bond. In addition, two protons at *δ*_H_ 3.59 and 3.27 suggested the presence of one oxygenated methylene group. ^13^C-NMR displayed 29 carbon resonances (Table 1), which could be classified as 2 CH_3_, 7 CH_2_, 15 CH and 5 non-protonated carbons by the DEPT method and HSQC data. Of them, one phenyl group and three double bonds could be readily identified, whilst the oxygenated methylene group could be observed at *δ*_C_ 65.9 (t). A preliminary analysis of these data suggested that **1** should be a cytochalasan product with a structure identical to that of cytochalasin B [18], a known cytochalasan isolated in this study. A further analysis of the NMR data suggested that **1** should be an isomer of cytochalasin B, with the loss of the hydroxy group at C-20 and the presence of oxygenated CH_2_ at *δ*_C_ 65.9 (t). The HMBC correlations from *δ*_H_ 3.59 and 3.27 (each 1H) with *δ*_C_ 42.4 (d, C-16), 37.5 (t, C-15) and 30.7 (t, C-17) revealed that the oxygenated CH_2_ should be placed at C-24 (Figure 2). It was supported by the ^1^H-^1^H COSY data, as shown in Figure 2. A detailed analysis of the ^1^H-^1^H COSY and HMBC data suggested that the other parts of **1** were the same as those of cytochalasin B (Figure 2). In the ROESY spectrum, the cross peaks of H-4/H-5, H-4/H-8 and H-7/H-13 suggested that **1** should have the same stereochemistry as cytochalasin B (Figure 3). Fortunately, the single crystal X-ray diffraction not only established the planar structure, but also determined the absolute configuration of **1**, as shown in Figure 4 (Flack parameter = 0.08(4); CCDC: 2240408). Finally, the structure of **1** was established and trivially named aspergicytochalasin A.

Compound **2** was isolated as a colorless oil. Its molecular formula was identified as C_29_H_35_NO_6_ by HRESIMS (measured at *m*/*z* 518.25122 [M + Na]^+^; calcd for C_29_H_37_NO_6_Na^+^: 518.25131). All the ^1^H- and ^13^C-NMR data (Table 1) of **2** showed similarities to those of **1** and cytochalasin B. A further analysis of the NMR data suggested that **2** should be 24-hydroxy-cytochalasin, as supported by the HMBC correlations from *δ*_H_ 3.60 and 3.26 (each 1H, H_2_-24) with *δ*_C_ 42.6 (d, C-16), 37.6 (t, C-15) and 31.1 (t, C-17) (Figure 2). It was further supported by the ^1^H-^1^H COSY data, as shown in Figure 2. Therefore, the structure of compound **2** was established and named aspergicytochalasin B.

Compound **3** was isolated as a colorless oil. Its molecular formula was identified as C_29_H_35_NO_6_ by HRESIMS (measured at *m*/*z* 516.23566 [M + Na]^+^; calcd for C_29_H_35_NO_6_Na^+^: 516.23566). The analysis of the ^1^H- and ^13^C-NMR data (Table 1) suggested that **3** should also be a cytochalasan, possessing a structure with a high similarity to that of cytochalasin B. However, a carbonyl carbon at *δ*_C_ 205.2 (s, C-15) suggested that there was one keto group in **3**. In the HMBC spectrum, the correlations from *δ*_H_ 0.99 (3H, d, *J* = 6.6 Hz, H_3_-24), 2.68 (1H, dd, *J* = 12.7 and 6.6 Hz, H-16) and 6.32 (1H, d, *J* = 15.3 Hz, H-14) with *δ*_C_ 205.2 indicated that the keto group should be placed at C-15, which allowed H-14 to be a doublet in the ^1^H-NMR spectrum (Figure 2). A comprehensive analysis of the HMBC, ^1^H-^1^H COSY and ROESY data suggested that the other parts of **3** were the same as those of cytochalasin B. Therefore, the structure of compound **3** was constructed and named aspergicytochalasin C.

Compound **4** was isolated as a colorless oil. The molecular formula of C_29_H_37_NO_6_ was determined based on HRESIMS (measured at *m*/*z* 518.25116 [M + Na]^+^; calcd for C_29_H_37_NO_6_Na^+^: 518.25131), corresponding with 12 degrees of unsaturation. Similarly, the ^1^H- and ^13^C-NMR data (Table 2) of **4** displayed high similarities with those of cytochalasin B. The main difference was that one non-protonated carbon at *δ*_C_ 74.3 (s, C-5) appeared in **4**. These data, as well as the analysis of the MS data, suggested that **4** should be a hydroxy product of cytochalasin B. In the HMBC spectrum, the correlations between *δ*_H_ 1.34 (3H, s, H_3_-11) and *δ*_C_ 74.3 (s), 55.6 (d, C-4) and 152.6 (s, C-6) supported that the OH should be placed at C-5 (Figure 2). A detailed analysis of the ^1^H-^1^H COSY and HMBC data suggested that the other parts of **3** were the same as those of cytochalasin B (Figure 2). In the ROESY spectrum, cross peaks of H_3_-11/H-3 and H_3_-11/H-7 were observed (Figure 3), which indicated that the OH at C-5 should be β-oriented. Based on the same biogenetic pathway for cytochalasin B and **4**, the absolute configuration of **4** was established, as shown in Figure 1. Finally, the structure of **4** was figured out and named aspergicytochalasin D.

Compound **5** was isolated as a colorless oil. According to HRESIMS (measured at *m*/*z* 496.26947 [M + H]^+^; calcd for C_29_H_38_NO_6_^+^: 496.26936), the molecular formula was identified as C_29_H_37_NO_6_, indicating 12 degrees of unsaturation. The ^1^H- and ^13^C-NMR data (Table 2) of **5** were closely related to those of cytochalasin B. A further analysis of these data suggested that **4** should have one additional OH group at the phenyl group, with respect to that of cytochalasin B. Four aromatic protons at *δ*_H_ 6.53 (1H, d, *J* = 1.8 Hz, H-2ʹ), 6.57 (1H, dd, *J* = 7.8 and 1.8 Hz, H-4ʹ), 6.55 (1H, d, *J* = 7.8 Hz, H-6ʹ) and 7.03 (1H, t, *J* = 7.8 Hz, H-5ʹ) could readily identify a 1,3-disubstituted phenyl moiety. The HMBC correlations from H-2ʹ to C-4ʹ and C-6ʹ further supported that the OH should be placed at C-3ʹ in **5**. A comprehensive analysis of the HMBC, ^1^H-^1^H COSY and ROESY data suggested that the other parts of **5** should be the same as those of cytochalasin B. Therefore, compound **5** was identified as 3ʹ-hydroxy-cytochalasin B and trivially named aspergicytochalasin E.

Compound **6** was isolated as a colorless oil. Its molecular formula was identified as C_29_H_39_NO_6_ by HRESIMS (measured at *m*/*z* 520.26679 [M + Na]^+^; calcd for C_29_H_39_NO_6_Na^+^: 520.26696), corresponding with 11 degrees of unsaturation. The ^1^H- and ^13^C-NMR data (Table 2) of **6** were similar to those of cytochalasin B. An analysis of these data indicated that the double bond between C-6 and C-12 in cytochalasin B had disappeared, instead of a methyl group at C-12 (*δ*_H_ 1.18; *δ*_C_ 22.3) and one non-protonated carbon at C-6 (*δ*_C_ 77.4) in **6**. It was supported by the key HMBC correlations from *δ*_H_ 1.18 (3H, s, H_3_-12) with *δ*_C_ 77.4 (s, C-6), 38.1 (d, C-5) and 76.3 (d, C-7) as well as the HMBC correlations from H-5 and H-7 with C-6 (Figure 2). A comprehensive analysis of the HMBC and ^1^H-^1^H COSY data suggested that the other parts of **5** were the same as those of cytochalasin B (Figure 2). In the ROESY spectrum, the cross peaks of H_3_-12/H-4 and H_3_-12/H-5 suggested that the OH at C-6 should be α-oriented, with respect to that of cytochalasin B (Figure 3). Therefore, the structure of **6** was established as 6-hydroxy-cytochalasin B and trivially named aspergicytochalasin F.

The structural characteristics of compounds **1**–**6** showed that they were similar to cytochalasin B, which was one of a large number of compounds found in this study. These results indicated that compounds **1**–**6** were likely to be obtained by the oxidation of cytochalasin B via p450, so the absolute configurations of compounds **1**–**6** should be consistent with that of cytochalasin B. The single crystal X-ray diffraction data for **1** also supported this conclusion.

In addition to the new compounds, the five known cytochalasans were identified as cytochalasin A (**7**) [18], cytochalasin B (**8**) [18], dihydrocytochalasin B (**9**) [19], cytochalasin Z2 (**10**) [20] and cytochalasin Z4 (**11**) [18] by a comparison of the spectroscopic data with those reported in the literature.

Compounds **1**–**6** were evaluated for their antibacterial activities against *Staphylococcus aureus*. Compounds **3** and **4** showed certain inhibitory activities, with MIC values of 128 and 64 μg/mL, respectively. In addition, compounds **1**–**6** were evaluated for their anti-inflammatory activities by inhibiting NO production in LPS-induced RAW 264.7 macrophages. As a result, compounds **1**, **3**, **5** and **6** showed inhibitory activities on NO production, with IC_50_ values less than 40 μM (Table 3).

## 4. Conclusions

In summary, a total of eleven cytochalasans, including six new ones, were isolated from the honeysuckle endophytic fungus *Aspergillus* sp. LE2. The new compounds exhibited antibacterial and anti-inflammatory activities. Cytochalasans are a class of important natural products with a rich structural diversity and biological activity [21,22,23,24,25,26]. At the same time, cytochalasans can also reflect the chemical relationship between microorganisms and hosts, and are an important bridge molecule in the study of medicinal plants and their functions. To the best of our knowledge, this is the first chemical study on honeysuckle endophytic fungi. Therefore, the new cytochalasans found in this study can provide certain information for further investigations on this plant and its endophytic fungal system.

## Figures and Tables

**Figure 1 jof-09-00374-f001:**
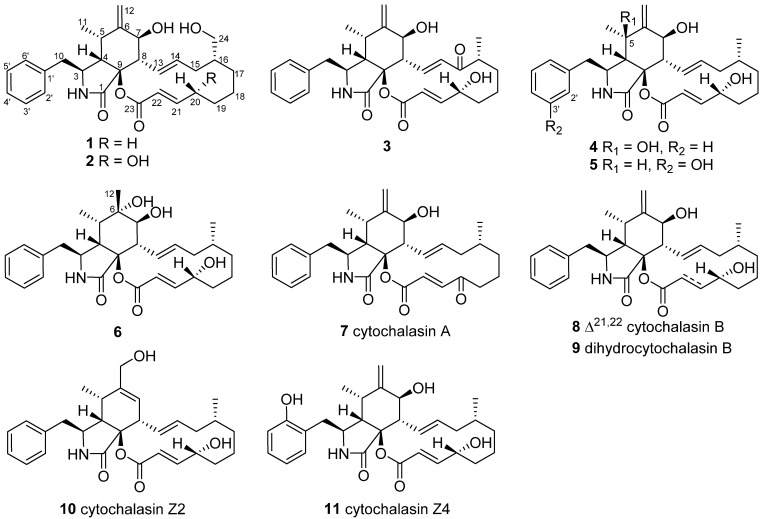
Structures of chemicals **1**−**11**.

**Figure 2 jof-09-00374-f002:**
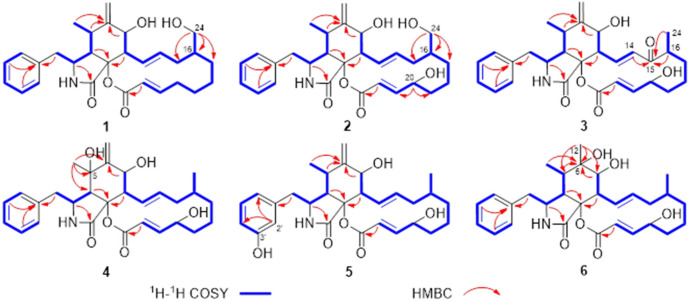
Key 2D NMR correlations of **1**–**6**.

**Figure 3 jof-09-00374-f003:**
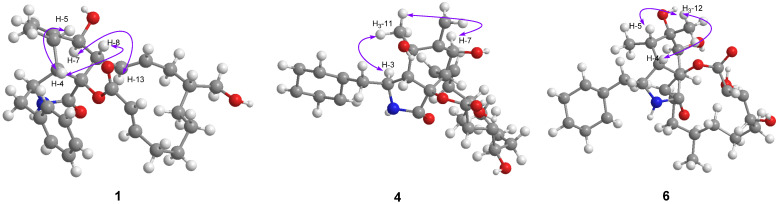
Key ROESY correlations of **1**, **4** and **6**.

**Figure 4 jof-09-00374-f004:**
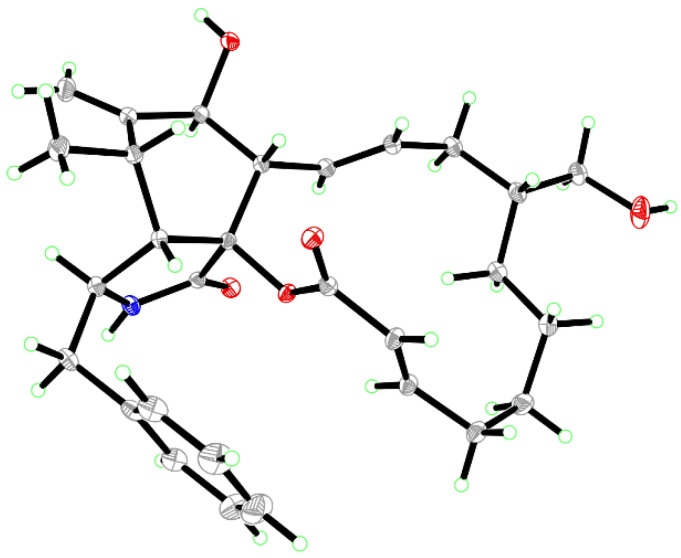
ORTEP diagram of **1** showing absolute configurations.

**Table 1 jof-09-00374-t001:** ^1^H (600 MHz) and ^13^C (150 MHz) NMR spectroscopic data for **1**–**3** in methanol-*d*_4_.

No.	1		2		3	
	*δ* _C_	*δ*_H_ (*J* in Hz)	*δ* _C_	*δ*_H_ (*J* in Hz)	*δ* _C_	*δ*_H_ (*J* in Hz)
1	173.9, C		174.0, C		173.0, C	
3	55.0, CH	3.36, m	54.8, CH	3.40, m	55.3, CH	3.40, td (6.1, 2.8)
4	48.9, CH	2.80, m	48.5, CH	2.85, dd (5.0, 2.4)	48.9, CH	2.84, m
5	32.9, CH	3.15, m	32.8, CH	3.21, m	33.1, CH	3.17, m
6	151.5, C		151.4, C		151.1, C	
7	71.1, CH	3.80, d (11.1)	71.5, CH	3.80, d (10.8)	70.8, CH	4.02, d (11.1)
8	49.9, CH	3.30, dd (11.1, 10.1)	49.2, CH	3.40, dd (10.8, 10.1)	49.0, CH	3.47, dd (11.1, 9.9)
9	84.9, C		85.4, C		86.1, C	
10	44.0, CH_2_	2.84, m 2.84, m	44, CH_2_	2.80, m	43.9, CH_2_	2.86, m
11	14.3, CH_3_	0.83, d (6.7)	14.1, CH_3_	0.85, d (6.7)	14.3, CH_3_	0.88, d (6.7)
12	114.3, CH_2_	5.08, s 5.30, s	114.4, CH_2_	5.28, s; 5.09, s	114.6, CH_2_	5.35, s; 5.13, s
13	129.1, CH	5.86, dd (15.0, 10.1)	129.2, CH	5.88, dd (14.7, 10.1)	145, CH	7.09, dd (15.3, 9.9)
14	136.5, CH	5.29, m	136.5, CH	5.24, m	134.5, CH	6.32, d (15.3)
15	37.5, CH_2_	2.40, m; 1.69, m	37.6, CH_2_	2.38, m; 1.66, m	205.2, C	
16	42.4, CH	1.35, m	42.6, CH	1.29, m	46.0, CH	2.68, dd (12.7, 6.6)
17	30.7, CH_2_	1.60, m; 0.86, m	31.1, CH_2_	1.53, m; 0.84, m	35.1, CH_2_	1.75, m; 1.29, m
18	27.6, CH_2_	1.70, m; 1.07, m	21.2, CH_2_	1.47, m; 1.29, m	22.0, CH_2_	1.48, m; 1.13, m
19	27.3, CH_2_	1.81, m; 1.35, m	35.4, CH_2_	1.92, m; 1.53, m	36.2, CH_2_	1.74, m; 1.66, m
20	35.2, CH_2_	2.37, m; 2.18, m	70.9, CH	4.46, m	71.5, CH	4.25, m
21	154.0, CH	7.02, m	154.4, CH	6.94, dd (15.6, 4.3)	153.1, CH	6.91, dd (15.7, 6.6)
22	122.0, CH	5.63, d (15.5)	119.6, CH	5.79, dd (15.6)	121.7, CH	5.84, d (15.7)
23	166.2, C		166.3, C		166.4, C	
24	65.9, CH_2_	3.59, dd (10.8, 3.6) 3.27, dd (10.8, 7.5)	65.9, CH_2_	3.60, dd (10.9, 3.4); 3.26, dd (10.9, 7.4)	15.1, CH_3_	0.99, d (6.6)
1′	138.5, C		138.2, C		138.4, C	
2′,6′	130.9, CH	7.14, d (7.4)	131.0, CH	7.13, d (7.4)	131.0, CH	7.15, d (7.3)
3′,5′	129.6, CH	7.26, t (7.4)	129.6, CH	7.27, t (7.4)	129.6, CH	7.27, t (7.3)
4′	127.8, CH	7.18, t (7.4)	127.9, CH	7.19, t (7.4)	127.9, CH	7.23, t (7.3)

**Table 2 jof-09-00374-t002:** ^1^H (600 MHz) and ^13^C (150 MHz) NMR spectroscopic data for **4**–**6** in methanol-*d*_4_.

No.	4		5		6	
	*δ* _C_	*δ*_H_ (*J* in Hz)	*δ* _C_	*δ*_H_ (*J* in Hz)	*δ* _C_	*δ*_H_ (*J* in Hz)
1	173.3, C		174.0, C		173.7, C	
3	57.2, CH	3.25, m	54.8, CH	3.32, m	55.0, CH	4.18, m
4	55.6, CH	2.86, d (4.2)	48.6, CH	2.81, dd (5.2, 2.6)	50.1, CH	2.58, m
5	74.3, C		32.9, CH	3.16, m	38.1, CH	2.27, m
6	152.6, C		151.5, C		77.4, C	
7	69.2, CH	3.94, d (12.2)	71.5, CH	3.76, d (10.9)	76.3, CH	3.45, d (12.3)
8	51.3, CH	3.83, dd (12.2, 10.2)	49.3, CH	3.34, dd (10.9, 9.8)	49.4, CH	2.86, dd (12.3, 10.2)
9	85.2, C		85.4, C		86.7, C	
10	44.0, CH_2_	3.02, d (6.6)	44.1, CH_2_	2.68, m	45.0, CH_2_	2.79, dd (13.7, 7.9) 2.93, dd (13.7, 4.4)
11	26.5, CH_3_	1.34, s	14.1, CH_3_	0.84, d (6.5)	13.0, CH_3_	0.99, d (7.1)
12	114.4, CH_2_	5.51, s; 5.47, s	114.3, CH_2_	5.24, s; 5.05, s	22.3, CH_3_	1.18, s
13	128.8, CH	5.62, dd (14.2, 10.2)	128.7, CH	5.79, dd (15.0, 9.8)	128.0, CH	5.64, dd (14.4, 10.2)
14	137.1, CH	5.41, m	136.7, CH	5.20, m	136.5, CH	5.25, m
15	42.7, CH_2_	2.19, m; 1.73, m	43.2, CH_2_	2.06, m; 1.61, m	42.8, CH_2_	2.14, m; 1.70, m
16	33.9, CH	1.46, m	34.5, CH	1.18, m	34.3, CH	1.35, m
17	35.7, CH_2_	1.83, m; 0.65, m	36.6, CH_2_	1.68, m; 0.53, m	36.1, CH_2_	1.77, m; 0.62, m
18	21.8, CH_2_	1.53, m; 1.42, m	21.5, CH_2_	1.41, m; 1.23, m	21.7, CH_2_	1.41, m
19	35.3, CH_2_	1.85, m; 1.61, m	35.4, CH_2_	1.87, m; 1.52, m	35.4, CH_2_	1.86, m; 1.58, m
20	72.8, CH	4.29, m	71.2, CH	4.42, m	72.0, CH	4.34, m
21	153.7, CH	7.04, dd (15.6, 6.3)	154.6, CH	6.92, dd (15.7, 5.4)	154.1, CH	6.99, dd (15.6, 6.4)
22	121.2, CH	5.79, d (15.6)	119.4, CH	5.74, d (15.7)	120.3, CH	5.75, d (15.6)
23	166.5, C		166.4, C		166.3, C	
24	20.7, CH_3_	0.90, d (6.5)	20.7, CH_3_	0.84, d (6.5)	20.7, CH_3_	0.88, d (6.6)
1′	139.2, C		139.7, C		139.0, C	
2′	130.5, CH	7.20, d (7.5)	117.8, CH	6.53, (1.8)	130.7, CH	7.19, d (7.4)
3′	129.8, CH	7.30, t (7.5)	158.5, C		129.6, CH	7.27, t (7.4)
4′	127.9, CH	7.22, t (7.5)	114.9, CH	6.57, dd (7.8, 1.8)	127.7, CH	7.19, t (7.4)
5′	129.8, CH	7.30, t (7.5)	130.5, CH	7.03, t (7.8)	129.6, CH	7.27, t (7.4)
6′	130.5, CH	7.20, d (7.5)	122.2, CH	6.55, d (7.8)	130.7, CH	7.19, d (7.4)

**Table 3 jof-09-00374-t003:** NO production inhibitions of Compounds **1**–**6**.

Entry	IC_50_ (μM)
**1**	26.8 ± 2.69
**2**	>40
**3**	17.1 ± 1.27
**4**	>40
**5**	31.2 ± 1.02
**6**	12.7 ± 1.85
SMS	7.3 ± 0.34

## Data Availability

The supporting information of this study is available from the corresponding author (Y.-C.L.). The CIF of **1** is available free of charge at https://www.ccdc.cam.ac.uk.

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
