# Peer review of "Cytochalasans from the Endophytic Fungus Aspergillus sp. LE2: Their Structures, Antibacterial and NO Production Inhibitory Activities"

_jof, 2023, doi:10.3390/jof9030374_

Round 1
Reviewer 1 Report
In this paper, seven new cytochalasans (1–6) were identified from endophyte Aspergillus sp. from the medicinal plant Lonicera japonica, together with a series of known compounds. The new structures were well determined via spectroscopic methods and aided by single crystal X-ray diffraction analysis. Some of the co-isolates showed antibacterial activities against Staphylococcus aureus and inhibitory activities on NO production. The structural elucidation part of the manuscript is sound and the conclusion are well drawn. In my opinion, it is suitable for rapid communication in the journal after minor revision.
The details of the IC50 data should be included in Abstract.
The SD values has to be added in the text.
In Supporting Information, the UV and IR spectra must be provided, and the structures should be pasted along all NMR spectra.
Add some of the recent literatures about cytochalasans, such as Angew. Chem. Int. Ed. 2016; 47: 3486 etc.
Author Response
Dear the Reviewer,
Thank you for the kind suggestions on our manuscript. We have revised them accordingly.
The details of the IC50 data should be included in Abstract.
Re. We have added them in the Abstract.
The SD values has to be added in the text.
Re. We have added them in the table.
In Supporting Information, the UV and IR spectra must be provided, and the structures should be pasted along all NMR spectra.
Re. We have made related revisions in Supporting Information.
Add some of the recent literatures about cytochalasans, such as Angew. Chem. Int. Ed. 2016; 47: 3486 etc.
Re. We have added six references in Conclusion.
Sincerely,
Yongchao Li
Reviewer 2 Report
The manuscript reports a series cytochalasans from endophytic fungus Aspergillus sp. . It introduced an interesting background for the exploration of the medicinal plant Lonicera japonica. The new structures are well established and the text is well organized and written. I believe it is worthy of publication on the Journal of Fungi. A question personally, why the authors choose rice as the cultrual medinum?
A minor revision is that "Seven" should be "Six" in Abstract
Overall, the manuscript is well established that I can not give more suggestions.
Author Response
We thank the Reviewer for the kind comments.
It is asked why the rice was set as cultural medium.
Actually, we tried many culture media including PDB, wheet, corn, and rice. The results indicated that the metabolites in rice medium are more abundant than in others. Therefore, we selected rice as the culture medinum.
In addition, we corrected "Seven" to "Six"
Many thanks for the suggestions.
Yongchao LI

Reviewer 3 Report
The paper plans to report new cytochalasans from the endophytic fungus Aspergillus sp. that isolated from the medicinal plant Lonicera japonica. Cytochalasans are a class of interesting fungal natural products enjoying a good reputation in the field. The structures are well elucidated by means of spectroscopic methods and X-ray diffraction. In addition, the text is well written which meets the requirements of the journal. I suggested for publication in its current form. By the way, the authors should change "Seven" to "Six" in the abstract when proofing the text.
Author Response
Many thanks for the kind comments.
We have revised "Seven" to "Six".

Reviewer 4 Report
Table 3: SD values are missing.
Table 2 should be placed before paragraph 2.4.
There is no discussion about the observed inhibitory effects on bacterial growth and NO production.
Conclusion section may be improved proposing further applications and/or studies.
Author Response
Dear the Reviewer,
Thank you for the kind comments.
We have revised the manuscript accordingly.
Table 3: SD values are missing.
Re. We have added them.
Table 2 should be placed before paragraph 2.4.
Re. Thank you for the suggestion. In order to keep the table in one page, we didn’t move the table 2, since there is not enough space for Table 2 before paragraph 2.4.
There is no discussion about the observed inhibitory effects on bacterial growth and NO production.
Re. These assays were carried out by Kunming Institute of Botany, Chinese Academy of Sciences. We would like to give further bioassay on these products in the future.
Conclusion section may be improved proposing further applications and/or studies.
Re. Thank you. We have revised the conclusion.
